# A Silicone Resin Coating with Water-Repellency and Anti-Fouling Properties for Wood Protection

**DOI:** 10.3390/polym14153062

**Published:** 2022-07-28

**Authors:** Zehao Ding, Wensheng Lin, Wenbin Yang, Hanxian Chen, Xinxiang Zhang

**Affiliations:** College of Materials Engineering, Fujian Agriculture and Forestry University, Fuzhou 350108, China; 15396095156@163.com (Z.D.); wensheng0817@163.com (W.L.); fafuywb@163.com (W.Y.)

**Keywords:** wood, superhydrophobic, hydrofluorosilicone oil (HFSO), silicone resin, anti-fouling

## Abstract

The strong hygroscopicity of wood greatly shortens its service life. Here, a simple impregnation modification approach was used to construct superhydrophobic silicone resin coatings on wood surfaces. Briefly, with hydrofluorosilicone oil (HFSO), tetramethyl tetravinyl cyclotetrasiloxane (V_4_), and hydrophobic SiO_2_ from industrial production as raw materials, superhydrophobic wood samples (water contact angle ~160.8°, sliding angle ~3.6°) can be obtained by simply dipping the wood in the HFSO/V_4_/SiO_2_ modifier solutions. As a result, the superhydrophobic silicone resin coating constructed on the wood surface still has good water repellency after finger touching, tape peeling, and sandpaper abrasion. When the mass ratio of HFSO to V_4_ is 2:1, the water absorption of the resulting wood after soaking in water for 24 h is only 29.2%. Further, the resulting superhydrophobic wood shows excellent anti-fouling properties. Finally, we believe that the impregnation modification method proposed in this study can be applied to the protection of cellulose substrates.

## 1. Introduction

Wood is a natural three-dimensional polymer composite material mainly composed of cellulose, hemicellulose, and lignin, which is easy to process and has good mechanical and environmental properties [1,2,3]. However, wood is a porous material and contains a large number of hydrophilic groups. It is prone to deformation, cracking, mildew, decay and degradation when placed in a humid environment for a long time [4,5,6,7]. Therefore, we need to protect the wood to inhibit or slow down the impact of moisture on the wood and prolong the service life of the wood. At present, scholars mainly build superhydrophobic coatings on wood surfaces through various modification methods to slow down the impact of moisture on wood [8,9,10]. Moreover, common hydrophobic coatings for wood protection are mainly silicone rubber coatings, silicone resin coatings, fluorosilane and nanomaterials, polysiloxane and nanomaterials, and so on. The construction of superhydrophobic coatings on wood substrate surfaces is also considered a promising approach.

Superhydrophobic treatment of wood can improve its dimensional stability, anti-mildew, anti-corrosion and anti-fouling, etc., and broaden the application field of wood [11,12,13,14,15]. The primary conditions for constructing a superhydrophobic surface on the wood surface are constructing a micro-nano hierarchical rough structure on the wood surface, followed by further modification with low surface energy materials to obtain superhydrophobic wood [16,17]. Thus far, the main methods for constructing superhydrophobic coatings on wood surfaces are: plasma treatment, sol-gel process, chemical vapor deposition, layer-by-layer assembly, and hydrothermal method [1,18,19,20,21,22,23]. In recent years, many studies have reported the successful preparation of durable superhydrophobic coatings on the wood surfaces. For example, Yang et al. proposed a novel method to synthesize robust superhydrophobic transparent coating on wood substrate using a multi-solvent continuous modification method. The obtained biomimetic wood showed excellent durability under mechanical shock and different temperature conditions [24]. Jia et al. prepared a high-wear-resistance superhydrophobic wood by an alkali-driven method. SiO_2_ nanoparticles and vinyltriethoxysilane (VTES) were used as raw materials. The resulting wood showed good chemical stability and mechanical stability [16]. Yang et al., using polydopamine (PDA), 3-mercaptopropyltriethoxysilane (KH580), Al_2_O_3_, and octadecyltrichlorosilane (OTS) as the raw materials, successfully fabricated superhydrophobic coatings on wood surface. The resulting superhydrophobic wood surface showed good resistance to organic solvents, acid and base corrosion, and flow scouring [25]. Obviously, although the above modification methods succeeded in constructing wear-resistant superhydrophobic coatings on wood surfaces, their preparation processes are cumbersome and mostly use expensive chemical reagents. Therefore, we are looking for a simple and effective method to construct superhydrophobic coatings on wood surfaces.

In this study, we constructed a superhydrophobic silicone resin coating with anti-fouling properties on wood surfaces by a simple one-step impregnation method. Briefly, hydrofluorosilicone oil (HFSO), tetramethyl tetravinyl cyclotetrasiloxane (V_4_), and hydrophobic SiO_2_ nanoparticles were chosen from industrial raw materials to construct superhydrophobic silicone resin coatings on wood substrates. Among them, HFSO plays a key role in the formation of superhydrophobic silicone resin coatings on wood surfaces. The reactive –Si–H bonds on its chains can undergo dehydrogenation and addition reactions with the –HC=CH_2_ groups in V_4_ and the –OH groups on the wood surface, respectively. Based on the reactivity of HFSO, a superhydrophobic silicone resin coating can be constructed on the wood surface by simply dipping the wood in the HFSO/V_4_/SiO_2_ modifier solutions. The wood will undergo dehydrogenation reaction with HFSO during the impregnation process, and will also undergo dehydrogenation reaction with the residual –Si–H bonds in the silicone resin formed by HFSO and V_4_. As a result, the obtained superhydrophobic wood still has good water repellency after abrasion with sandpaper, finger touch, tape peeling, etc. Moreover, the results of this study are expected to be used in the protection of various hydroxyl-containing materials.

## 2. Materials and Methods

### 2.1. Materials

The wood samples (Chinese fir) with a density of 338 ± 37 kg/m^3^ were provided by Furen Wood (Fuzhou) Co., Ltd. (Fujian, China), and cut into pieces with dimensions of 20 mm × 20 mm × 10 mm for use. The wood samples were ultrasonically rinsed with deionized water for 1 h and then dried in an oven at 80 °C for 10 h. Hydrophobic SiO_2_ nanoparticles (80 nm) and tetramethyl tetravinyl cyclotetrasiloxane (V_4_) were obtained from Xiamen Guiyou New Material Technology Co., Ltd. (Fujian, China). Hydrofluorosilicone oil (HFSO) was purchased from Dongguan Hongya Silicone Co., Ltd. (Dongguan, China). N-hexane was purchased from Xilong Scientific Co., Ltd. (Guangdong, China). Karstedt’s catalyst was purchased from Shanghai Vivo Chemical Co., Ltd. (Shanghai, China).

### 2.2. Preparation of HFSO/V_4_/SiO_2_ Modifier Solutions

First, 1 g hydrophobic silica was dispersed in a beaker with 60 g of n-hexane. Then, the different mass ratios of HFSO and V_4_ (1:1, 1:2, 1:3, 2:1, 3:1) were added in the above solution. Finally, 0.2 mL of Karstedt’s catalyst was added to the HFSO/V_4_/SiO_2_ solution and the mixture was magnetically stirred at room temperature for 10 min to obtain HFSO/V_4_/SiO_2_ modifier solutions.

### 2.3. Preparation of Superhydrophobic Wood Samples

The unmodified wood samples were immersed in the HFSO/V_4_/SiO_2_ modifier solutions for 10 min, then taken out and air-dried for 1 min. After three cycles of dipping, the wood samples were placed in an oven at 105 °C for heat treatment for 1 h. In addition, by calculation, the consumption of modified solution was 1.49 kg/m^2^. Finally, the superhydrophobic wood samples with anti-fouling properties were obtained.

### 2.4. Characterization

Infrared spectrometer (FTIR, Bruker Tensor 27, Frankfurt, Germany) was carried out to analyze the surface chemistry change of wood samples. Test conditions: the spectral range was 400–4000 cm^−1^, the resolution was 4 cm^−1^, and the number of scans was 64. The surface morphology changes of wood were observed by Zeiss scanning electron microscope at an accelerating voltage of 20.0 kV (SEM, ZEISS Z500, Jena, Germany). Since the wood is not conductive, to better observe the change of the wood morphology, the wood needed to be sprayed with gold before the SEM test, and the gold spraying time was 100 s. The water contact angle (WCA) and sliding angle (SA) of wood before and after modification were carried out on a contact angle meter at room temperature using 5 μL and 10 μL of water as probe liquid (HARKE-SPCA-1, Beijing, China). More importantly, five points on each test surface were selected for testing and the average was taken to get the final WCA and SA results.

## 3. Results

### 3.1. Formation Mechanism of Superhydrophobic Silicone Resin Coating on Wood Surface

The formation mechanism of the superhydrophobic silicone resin coating on the wood surface is based on the dehydrogenation reaction of the residual –Si–H bonds in the silicone resin structure formed by the reaction of HFSO and V_4_ with the –OH groups on the wood surface, and then the silicone resin is covalently bonded on the wood surface and reduces the surface energy of the wood (seen in Appendix A). On the other hand, due to the introduction of hydrophobic SiO_2_ nanoparticles into the reaction system, the low surface energy combined with rough micro-nanostructures, therefore, the construction of superhydrophobic silicone resin coatings on wood surfaces can be achieved. As shown in Figure 1b, HFSO is a siloxane with a large number of –Si–H bonds and hydrophobic –CH_3_ groups on its chain, and a small number of –CH_2_CH_2_CF_3_ groups on its chain. Therefore, HFSO is a low surface energy material. In the presence of catalysts, the –Si–H bonds on the HFSO chain can undergo addition reactions and dehydrogenation reactions with the –CH=CH_2_ groups in the V_4_ structure (Figure 1a) and the –OH groups on the wood surface (Figure 1b). Among them, the addition reaction of HFSO and V_4_ forms a transparent silicone resin (Figure 1a), and the structure of the silicone resin contains unreacted –Si–H bonds. Based on the reaction characteristics of HFSO, we can construct a superhydrophobic silicone resin coating on the wood surface by simply dipping the wood in the modifier solutions of HFSO/V_4_/SiO_2_, as shown in Figure 1d. During the wood impregnation process, the low surface energy silicone resin formed by HFSO and V_4_ wrap the hydrophobic SiO_2_ nanoparticles and build up on the wood surface in the form of covalent bonds, thus constructing a layer of superhydrophobic silicone coating on the wood surface.

### 3.2. Effect of Mass Ratio of HFSO/V_4_ on the Hydrophobic Properties of Silicone Resin Coatings on Wood Surfaces

The mass ratio of HFSO/V_4_ has a certain influence on the hydrophobicity of the silicone resin coating formed on the wood surface. Through repeated experiments, it has been found that when the amount of HFSO or V_4_ is too large, the hydrophobicity of the silicone resin coating on the wood surface will decrease, as shown in Figure 2a. On the one hand, when there are more HFSO (HFSO/V_4_ 3:1), it will first dehydrogenate with the –OH groups on the wood surface, resulting in no sites on the wood surface and then react with the –Si–H bonds in the silicone resin structure. On the other hand, when there is more V_4_ (HFSO/V_4_ 1:3), the reaction between HFSO and V_4_ is relatively complete, and there are less residual –Si–H bonds in the formed silicone resin structure, resulting in less silicone resin grafted on the wood surface. Therefore, more HFSO and V_4_ will lead to a decrease in the hydrophobicity of wood. In this study, through repeated experiments, it is known that when the mass ratio of HFSO and V_4_ is 2:1, the hydrophobicity of the silicone resin coating on the modified wood surface is the best, and the WCA reaches 157.6° (Figure 2a_2_). In addition, we also tested the SA of cross sections of treated wood with different HFSO/V_4_ mass ratios. Obviously, the SA of the modified wood cross section is lower than 10°. These results indicate that changing the mass ratio of HFSO to V_4_ can realize the construction of superhydrophobic silicone resin coatings on wood surfaces.

### 3.3. FTIR Analysis

FTIR spectra of wood before and after superhydrophobic silicone resin coating modification were recorded and are presented in Figure 3. Obviously, some new absorption peaks appeared on the surface of the modified wood (Figure 3b). Among them, the absorption peaks at 2966, 1214, and 901 cm^−1^ were the stretching vibration of the –Si–CH_3_ groups from HFSO or V_4_ [12,26]. Bands at 2167 cm^−1^ corresponded to –Si–H of HFSO [17]. In addition, the absorption band at 1092, 840, and 472 cm^−1^ belonged to the Si–O–Si characteristic absorption bands from HFSO and bands at 958 cm^−1^ were attributed to –Si–OH of hydrophobic SiO_2_ [4,27]. Moreover, it can be seen from the FTIR test results that the absorption peak of –OH groups on the surface of the modified wood is weakened, which is mainly due to the dehydrogenation reaction between the –OH groups on the wood surface and the –Si–H bond in the silicone resin. Therefore, the FTIR test results can indicate the presence of a silicone resin coating on the treated wood surface.

### 3.4. Surface Morphology Analysis

To analyze the morphology difference of wood before and after modification, a SEM test was carried out. As shown in Figure 4a,b, the radial section of untreated wood contains many micron-sized pores, and the surface is relatively smooth, which makes it easy to absorb water and moisture from the environment and has poor hydrophobicity. For the wood samples treated with superhydrophobization (HFSO/V_4_ 2:1 treated wood), the surface roughness was significantly increased, and the micro-nano hierarchical rough structure appeared on the surface, so the hydrophobicity was significantly improved. In addition, we can see in the high-magnification images (Figure 4d) that the treated wood surface forms an island-like structure, which is a special structure formed by the silicone resin encapsulating the hydrophobic SiO_2_ nanoparticles. It is precisely because of the formation of this island-like structure that the treated wood has excellent water repellency. Therefore, combined with the results of FTIR and SEM test analysis, it can be seen that the formation of micro-nano hierarchical rough structure and the introduction of low surface energy hydrophobic materials jointly promote the construction of superhydrophobic silicone resin coating on the treated wood surface.

### 3.5. Evaluation of Hygroscopicity and Dimensional Stability

To evaluate the water repellency of the superhydrophobic silicone resin coating on the treated wood surface, we conducted 24 h water absorption and anti-swell efficiency (ASE) tests according to the previous research reports [28,29]. First, as shown in Figure 5a, the water absorption rate of unmodified wood after soaking in water for 24 h reached 161.3%, indicating that it has strong hygroscopicity. In contrast, the water absorption of wood coated with a superhydrophobic silicone resin coating decreased significantly after immersion in water for 24 h. Moreover, when the mass ratio of HFSO and V_4_ is 2:1, the water absorption rate of treated wood soaked in water for 24 h is only 29.2%, indicating that it has strong water repellency. In addition, it can be seen from the physical image (Figure 5a_1_) that the surface of the modified wood will be in a mirror-like shape when immersed in water, which further shows that the modified wood has good hydrophobicity. As shown in Figure 5b, except for the HFSO/V_4_ 3:1 treated wood samples, the ASE values of other modified woods all exceeded 60%, which means that the treated woods have better dimensional stability [29]. In conclusion, the 24-h hygroscopicity and ASE test results show that the silicone resin coating constructed on the modified wood surface has good water repellency.

### 3.6. Durability Analysis of Modified Wood

The durability and abrasion resistance of the coating are important factors for realizing the practical application of superhydrophobic wood. To investigate the durability of the superhydrophobic silicone resin coating, the treated wood was studied in a constant temperature and humidity chamber at a temperature of 50 °C and a relative humidity of 70%, as shown in Figure 6a. The treated wood samples were taken out every 8 h and then tested for WCA and SA. Obviously, the treated wood still had superhydrophobic properties (seen in Figure 6a) after being placed for 48 h. The WCA and SA were greater than 150° and less than 10°, respectively. The above results show that the superhydrophobic silicone resin coating on the modified wood surface has good durability against moisture attack. In addition, to further evaluate the durability of the superhydrophobic silicone resin coating, we also conducted a sandpaper abrasion resistance test. The superhydrophobic wood sample was placed on a sandpaper (1500 mesh), a weight of 20 g was placed on the sample, and then the sample was pushed to wear on the sandpaper. After every 10 cm of wear, the WCA was tested, as shown in Figure 6b. The results of repeated experiments show that the superhydrophobic silicone resin coating on the treated wood surface still maintains superhydrophobicity after being worn by sandpaper for 130 cm. This means that the superhydrophobic silicone resin coating has good abrasion resistance.

To further evaluate the abrasion resistance of the superhydrophobic silicone resin coating on the treated wood surface, finger touch and tape peel tests were performed on the superhydrophobic wood samples (HFSO/V_4_ 2:1 modified wood). As shown in Figure 7a,b, after the treated wood sample was touched and pressed by fingers (1 min), the water droplets can still be spherical on the wood surface, with a WCA of 154.3°. This result indicates that the superhydrophobic silicone resin coating on the treated wood surface has good resistance to finger touch and pressing. In addition, as shown in Figure 7c,d, raw tape was forcefully stuck on the surface of the superhydrophobic wood for 30 s and then torn off. The results show that the water droplets can still stand in the shape of spheres on the surface of the worn wood with a WCA of 155.6°. In conclusion, finger touch and tape peel tests show that the superhydrophobic silicone resin coating adheres well to the wood substrate. This is mainly attributed to the fact that the superhydrophobic silicone resin coating was covalently grafted on the wood surface.

### 3.7. Anti-Fouling Properties of Superhydrophobic Silicone Resin Coatings

Low adhesion of superhydrophobic coatings can give them anti-fouling properties [30,31]. In this study, we used common liquids such as cola, orange juice, and methyl blue droplets to evaluate the anti-fouling properties of superhydrophobic silicone resin coatings on treated wood surfaces. As shown in Figure 8, cola (Figure 8a_1_–a_3_), methyl blue (Figure 8b_1_–b_3_), and orange juice (Figure 8c_1_–c_3_) liquid was dropped on the surface of the treated wood using a plastic dropper. The results showed that the droplets could quickly roll off the surface of the treated wood in the shape of a sphere without sticking to it. As a comparison, the above liquids easily adhered to the surface of unmodified wood, causing serious contamination to it (seen in Appendix A). Based on the above experimental results, it can be confirmed that the superhydrophobic silicone resin coating on the treated wood surface has good anti-fouling performance.

## 4. Conclusions

In summary, we successfully synthesized a modifier solution composed of HFSO, V_4_, and hydrophobic SiO_2_ for constructing superhydrophobic silicone resin coatings on wood substrate surfaces. The silicone resin formed by HFSO and V_4_ can tightly anchor the hydrophobic SiO_2_ on the wood surface. This makes a superhydrophobic silicone resin coating formed on the treated wood surface with good abrasion resistance, including resistance to finger touch, tape peeling, and sandpaper abrasion. The treated wood samples still have superhydrophobic properties after being placed in a constant temperature and humidity chamber for 48 h. In addition, when the mass ratio of HFSO to V_4_ was 2:1, the water absorption and ASE of the superhydrophobic wood samples soaked in water for 24 h were 29.2% and 75%, respectively. This indicates that it has good moisture resistance. Last but not least, the treated wood samples also showed excellent anti-fouling properties to common liquids, including cola, orange juice, and methyl blue.

## Figures and Tables

**Figure 1 polymers-14-03062-f001:**
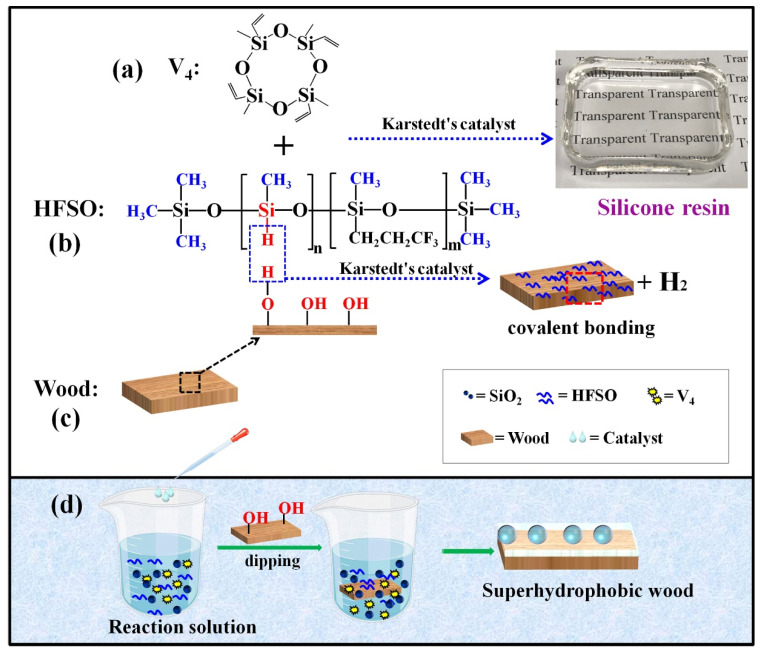
Schematic illustration of the functions and reactions of all components during the construction of superhydrophobic silicone resin coatings on wood surfaces: (**a**) V_4_ structure; (**b**) HFSO structure; (**c**) Wood; (**d**) Schematic diagram of wood modification.

**Figure 2 polymers-14-03062-f002:**
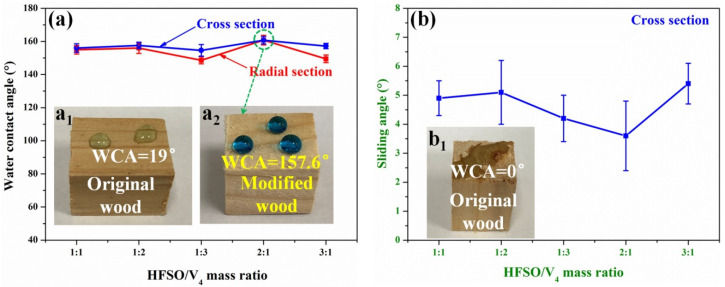
WCA (**a**) and SA (**b**) on wood surface at different HFSO/V_4_ mass ratio. (**a_1_**,**a_2_**,**b_1_**) Methyl blue water droplets on original and treated wood surfaces.

**Figure 3 polymers-14-03062-f003:**
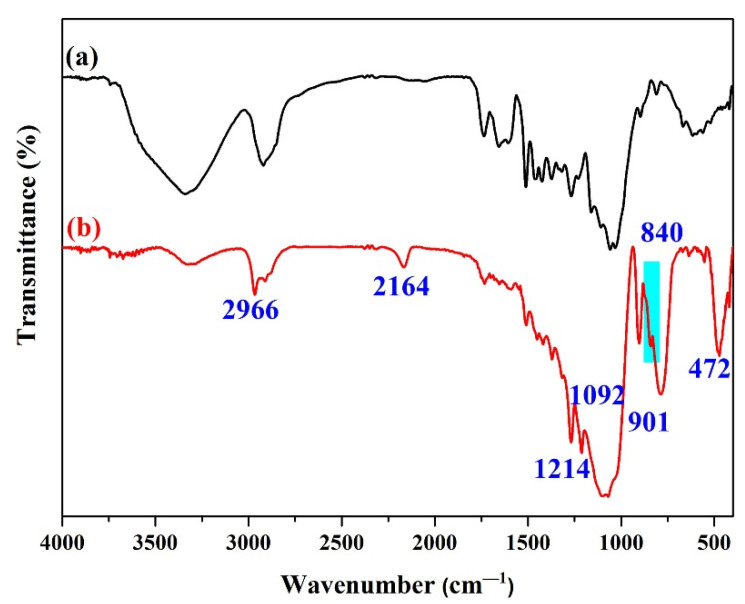
FTIR spectra of original wood (**a**) and HFSO/V_4_ 2:1 modified wood sample (**b**).

**Figure 4 polymers-14-03062-f004:**
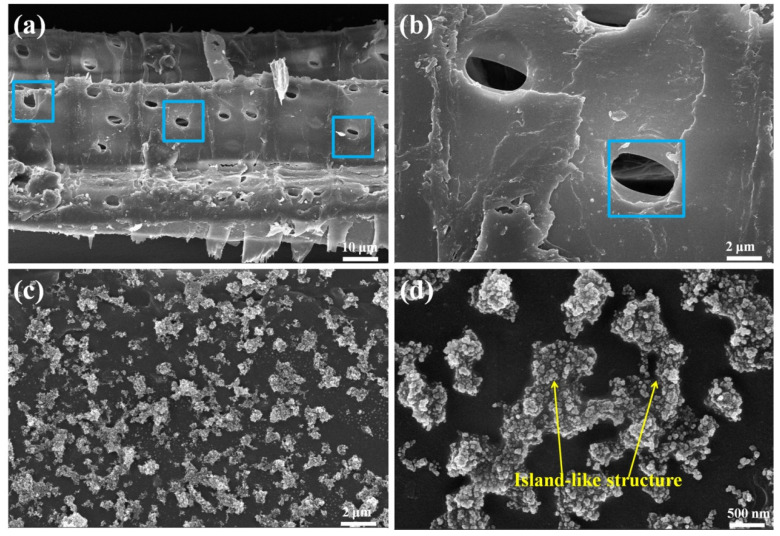
SEM images of original wood surface (**a**,**b**) and superhydrophobic silicone resin coatings on the treated wood surface (**c**,**d**).

**Figure 5 polymers-14-03062-f005:**
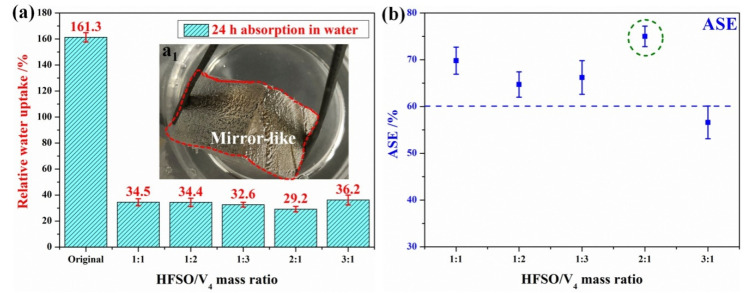
Test of water absorption (**a**) and anti-swell efficiency (**b**) of wood before and after modification after soaking in water for 24 h. (**a_1_**) image of wood soaked in water.

**Figure 6 polymers-14-03062-f006:**
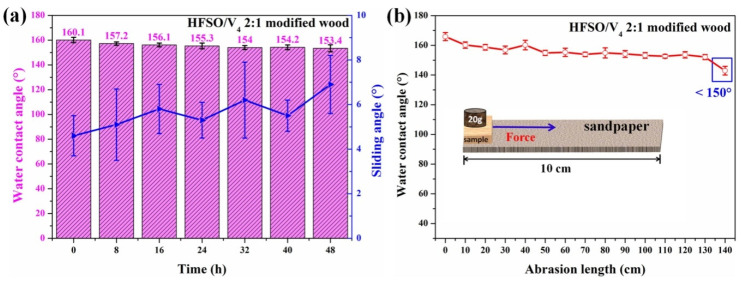
The durability test of superhydrophobic silicone resin coating on modified wood surface. (**a**) Variation of the WCA and SA under 70% relative humidity at 50 °C; (**b**) sandpaper abrasion test.

**Figure 7 polymers-14-03062-f007:**
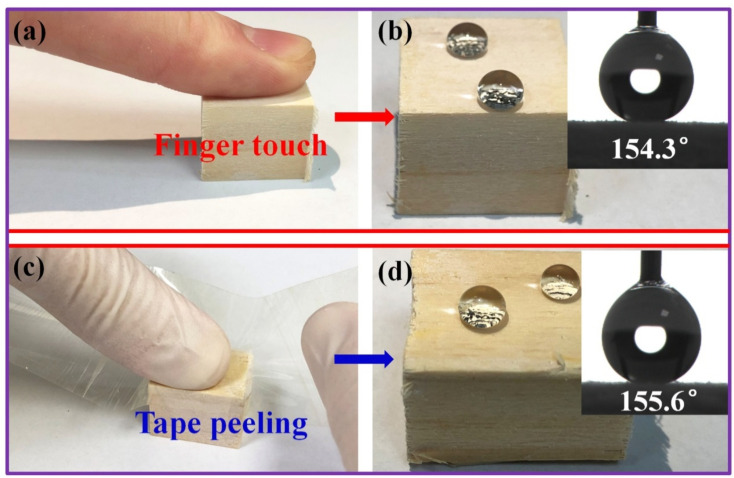
Finger touching (**a**,**b**) and tape peeling (**c**,**d**) tests on superhydrophobic silicone resin coating surface.

**Figure 8 polymers-14-03062-f008:**
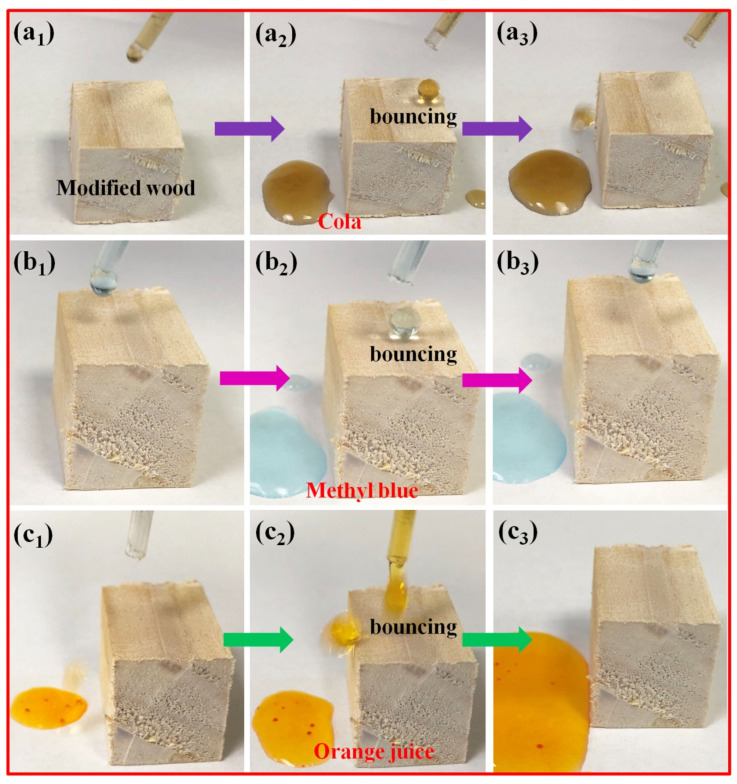
Anti-fouling performance test of superhydrophobic silicone resin coating on treated wood surface: (**a_1_**–**a_3_**) Wetting resistance test of superhydrophobic wood to cola; (**b_1_**–**b_3_**) Wetting resistance test of superhydrophobic wood to methyl blue; (**c_1_**–**c_3_**) Wetting resistance test of superhydrophobic wood to orange juice.

## Data Availability

The data presented in this study are available on request from thecorresponding author.

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
