# Peer review of "A Silicone Resin Coating with Water-Repellency and Anti-Fouling Properties for Wood Protection"

_polymers, 2022, doi:10.3390/polym14153062_

Round 1

Reviewer 1 Report

Zehao Ding et al considered the treatment of wood with silicone-containing substances to create a hydrophobic layer on the surface. Immediately after the beginning of acquaintance with the manuscript, the question arises - what is the cost of the reagents used? The internet price for them is tens and hundreds of dollars. It is very expensive and it is unlikely that at such prices the results presented in the manuscript will be widely used. Next, the authors must indicate the exact consumption of the substance per square meter, how many layers, drying time. The authors use the term emulsion in their manuscript, but the corresponding microphotographs are not given in the manuscript. It is not clear how the viscosity of the system changes with varying emulsion components and other rheological properties. And as you know, rheological properties are very important when using paints, emulsions, etc. At the moment, the manuscript is more suitable for a materials science journal than a scientific one. The authors should conduct a deeper theoretical analysis and highlight the scientific novelty of the work. The same goes for the conclusions of the manuscript. In my opinion, it is important to justify the choice of reagents used, showing all their pros and cons, including environmental features. When using liquid glasses, it is usually recommended to apply a primer. Is this procedure required in the opinion of the authors when using the described systems?   Line 46. "VTES" - needs to be decoded. Figure 2. Why do the authors use methyl blue water? The use of a dye may affect the observed result. Figure 4. Why does heterogeneous morphology form on the surface of the modified sample? What happens to micropores? Was there any pretreatment of the wood? This is important because, for example, in the work https://doi.org/10.3390/membranes12030297 it is shown that such treatments lead to a change in the pore system in the sample.

Author Response

Response to Reviewer 1 Comments

Point 1: What is the cost of the reagents used? The internet price for them is tens and hundreds of dollars. It is very expensive and it is unlikely that at such prices the results presented in the manuscript will be widely used.

Response 1: We deeply appreciate the reviewer's suggestion and comment. In fact, the total cost of the reagents used in this study is less than 20 RMB. All the raw materials we use have been industrialized. HFSO lower than 200 RMB/kg, V4 lower than 30~50 RMB/kg, hydrophobic SiO2 lower than 50 RMB/kg.

Point 2: Next, the authors must indicate the exact consumption of the substance per square meter, how many layers, drying time.

Response 2: Thank you very much for your suggestion and comment. We have corrected it in the new submitted manuscript. By calculation, the consumption of modified solution is 1487.5 g/m2. The number of layers is 3 and the drying time is 1 hour.

Point 3: The authors use the term emulsion in their manuscript, but the corresponding microphotographs are not given in the manuscript. It is not clear how the viscosity of the system changes with varying emulsion components and other rheological properties. And as you know, rheological properties are very important when using paints, emulsions, etc.

Response 3: Thank you very much for your suggestion and comment. After serious consideration, we think the expression of “emulsion” is wrong, and therefore we corrected the emulsion to a solution in the newly submitted manuscript.

Point 4: At the moment, the manuscript is more suitable for a materials science journal than a scientific one. The authors should conduct a deeper theoretical analysis and highlight the scientific novelty of the work. The same goes for the conclusions of the manuscript.

Response 4: Thank you very much for your suggestion. In fact, the journal Polymers has published numerous articles on superhydrophobic modification of wood. Therefore, we consider this study to be compliant with the journal of Polymers.

Point 5: In my opinion, it is important to justify the choice of reagents used, showing all their pros and cons, including environmental features.

Response 5: Thank you very much for your suggestion and comment. The modifier raw materials HFSO and V4 used in this study are all organosilicon materials, which are non-toxic and tasteless, and will not damage the environment. In addition, hydrophobic SiO2 nanoparticles are also nontoxic materials.

Point 6: When using liquid glasses, it is usually recommended to apply a primer. Is this procedure required in the opinion of the authors when using the described systems.

Response 6: We deeply appreciate the reviewer's suggestion and comment. In fact, the inorganic material used in this study is the industrially produced hydrophobic SiO2 nanoparticles, which can be well dispersed in the modifier solution. When the wood sample is dipped in the modification solution, the hydrophobic SiO2 is anchored to the wood surface by the silicone resin coating formed by the reaction of HFSO and V4. So I don't think a primer is needed when using hydrophobic SiO2 in this study

Point 7: Line 46. "VTES" - needs to be decoded.

Response 7: We deeply appreciate the reviewer's suggestion and comment. We have corrected it in the new submitted manuscript.

Point 8: Figure 2. Why do the authors use methyl blue water? The use of a dye may affect the observed result.

Response 8: Thank you very much for your suggestion. In fact, we refer to other people's research reports(Jia S, Liu M, Wu Y, et al. Facile and scalable preparation of highly wear-resistance superhydrophobic surface on wood substrates using silica nanoparticles modified by VTES. Applied Surface Science, 2016, 386: 115-124), (Wu Y, Jia S, Qing Y, et al. A versatile and efficient method to fabricate durable superhydrophobic surfaces on wood, lignocellulosic fiber, glass, and metal substrates. Journal of Materials Chemistry A, 2016, 4(37): 14111-14121), the water droplets were dyed with methyl blue to better demonstrate the water repellency of the modified wood. And the dyeing treatment will not affect the results.

Point 9: Figure 4. Why does heterogeneous morphology form on the surface of the modified sample?.

Response 9: Thank you very much for your suggestion and comment. The rough surface structure formed on the modified wood surface is caused by the encapsulation of hydrophobic SiO2 nanoparticles by silicone resin formed by the reaction of HFSO with V4.

Point 10: What happens to micropores? Was there any pretreatment of the wood? This is important because, for example, in the work https://doi.org/10.3390/membranes12030297 it is shown that such treatments lead to a change in the pore system in the sample.

Response 10: Thank you very much for your suggestion. The wood has a simple treatment before modification (the wood samples were ultrasonically rinsed with deionized water for 1 h and then dried in an oven at 80 °C for 10 h.), the purpose is to remove some stains left on the wood surface. In addition, we would like to point out that this study mainly focuses on the surface hydrophobization of wood, and does not study the changes in the internal structure and morphology of wood.

Reviewer 2 Report

The manuscript entitled "A Silicone Resin Coating with Water-repellency and Anti-fouling Properties for Wood Protection" by X. Zhang et al. presents some approaches to prepare a silicone resin coating in order to be used in wood protection applications. The idea of the paper is good, even some several reports on the utilization of HFSO, V4 and silica for wood protection are known and can be also added to references.

However, this paper also contains some new data that can be of interest for readers, but also I have some observations that can be addressed before publication:

- In Abstract, the authors should correct the scientific name of V4. In the last sentence the authors also have to remove "any hydroxyl group-containing substrates" with some general polar surfaces, not only hydroxyl groups presence could affect the hydrophylicity of a substrate.

Introduction - the authors should mention some hydrophobic coatings usually used, even commercial ones at references [8-10].

- there are also some abbreviations to be explained at lines 46-48, page 2.

- in the last paragraph, lines 56-69, it is not clear if the reaction between the wood substrate and coating composition occur in one step. Please be more explicit.

- Materials and methods: Karstedt's catalyst should be  correctly written and also added to Materials section.

- Results: the mechanism described in the first paragraph, lines 105-109,  should be schematic represented for a better understanding. Si-H groups already exists, these one are not formed by the reaction of HFSO and V4. Please correct the errors within this paragraph.

-in Figure 1, also correct the name of the catalyst.

-the authors also claimed the chemical bonds between the resin and wood surface, please describe with which component of the wood and why it is support the utilization of a catalyst, lines 139-141. Also these modification in wood surfaces must be confirmed by FTIR spectroscopy and SEM. As it can be observed in SEM images, the silicone coating is not a continuous film of the surface of wood?!

- The durability studies are only made over an 8-hour period. It too little, some artificial thermal aging studies would be welcome over a period of thousands of hours.

On the basis of these considerations regarding this paper, I recommend the publication after Minor revision.

Author Response

Response to Reviewer 2 Comments

We deeply appreciate the reviewer's suggestion and comment. Based on your comments and suggestions, we have made a lot of revisions in the new submitted manuscript, and the revised parts use red fonts.

Point 1: The manuscript entitled "A Silicone Resin Coating with Water-repellency and Anti-fouling Properties for Wood Protection" by X. Zhang et al. presents some approaches to prepare a silicone resin coating in order to be used in wood protection applications. The idea of the paper is good, even some several reports on the utilization of HFSO, V4 and silica for wood protection are known and can be also added to references.

Response 1: Thank you very much for your suggestion. In fact, there is almost no research on the use of HFSO, V4 for wood protection. But, the use of silica for wood protection has been well documented and we cite it in the manuscript.

Point 2: In Abstract, the authors should correct the scientific name of V4. In the last sentence the authors also have to remove "any hydroxyl group-containing substrates" with some general polar surfaces, not only hydroxyl groups presence could affect the hydrophylicity of a substrate.

Response 2: We deeply appreciate the reviewer's suggestion and comment. We have corrected it in the new submitted manuscript.

Point 3: Introduction - the authors should mention some hydrophobic coatings usually used, even commercial ones at references [8-10].

Response 3: We deeply appreciate the reviewer's suggestion and comment. We have corrected it in the new submitted manuscript.

Point 4: there are also some abbreviations to be explained at lines 46-48, page 2.

Response 4: Thank you very much for your suggestion. The new submitted manuscript was revised carefully.

Point 5: in the last paragraph, lines 56-69, it is not clear if the reaction between the wood substrate and coating composition occur in one step. Please be more explicit.

Response 5: Thank you very much for your suggestion. The new submitted manuscript was revised carefully.

Point 6: Materials and methods: Karstedt's catalyst should be correctly written and also added to Materials section.

Response 6: We deeply appreciate the reviewer's suggestion and comment. We have corrected it in the new submitted manuscript.

Point 7: Results: the mechanism described in the first paragraph, lines 105-109, should be schematic represented for a better understanding. Si-H groups already exists, these one are not formed by the reaction of HFSO and V4. Please correct the errors within this paragraph.

Response 7: Thank you very much for your suggestion. The new submitted manuscript was revised carefully.

Point 8: in Figure 1, also correct the name of the catalyst.

Response 8: Thank you very much for your suggestion. The new submitted manuscript was revised carefully.

Point 9: the authors also claimed the chemical bonds between the resin and wood surface, please describe with which component of the wood and why it is support the utilization of a catalyst, lines 139-141. Also these modification in wood surfaces must be confirmed by FTIR spectroscopy and SEM. As it can be observed in SEM images, the silicone coating is not a continuous film of the surface of wood?!.

Response 9: Thank you very much for your suggestion. In fact, we have stated in the manuscript that –Si-H bonds remain in the structure of the silicone resin produced by the reaction of HFSO with V4. In the presence of a Karstedt's catalyst, it can undergo a dehydrogenation reaction with the –OH components on the surface of the wood, thereby grafting the silicone resin on the surface of the wood. It can be seen from the FTIR test results that the absorption peak of –OH groups on the surface of the modified wood is weakened, which is mainly due to the dehydrogenation reaction between the -OH groups on the wood surface and the -Si-H bond in the silicone resin. In this experiment, the wood was immersed in the n-hexane modification solution containing HFSO, V4, and SiO2, and there was a dispersant, so it was impossible to form a continuous silicone resin layer on the surface of the wood.

Point 10: The durability studies are only made over an 8-hour period. It too little, some artificial thermal aging studies would be welcome over a period of thousands of hours.

Response 10: Thank you very much for your suggestion. In fact, The treated wood samples were taken out every 8 h and then tested for WCA and SA. Obviously, the treated wood still has superhydrophobic properties after being placed for 48 h. It is undeniable that the superhydrophobic wood we prepared could not stand up to thousands of hours of aging tests.

Point 11: On the basis of these considerations regarding this paper, I recommend the publication after Minor revision.

Response 11: Thank you very much for your suggestion. The new submitted manuscript was revised carefully.

Round 2

Reviewer 1 Report

Response 2: Thank you very much for your suggestion and comment. We have corrected it in the new submitted manuscript. By calculation, the consumption of modified solution is 1487.5 g/m2. The number of layers is 3 and the drying time is 1 hour. - It's better to use kg rather than gr.   Response 3: Thank you very much for your suggestion and comment. After serious consideration, we think the expression of “emulsion” is wrong, and therefore we corrected the emulsion to a solution in the newly submitted manuscript. - The authors should answer the question, what happens to the viscosity?!   Line 12. "composite solution" - do I understand correctly that the mixed solution contains a second phase? Or is it a true solution, then the term composite cannot be used? Line 162. "the physical image", recommend removing "physical" Line 183. "absorb" - need to check the term. Line 206. If the treated material has superhydrophobic properties, then what is the sorption of water (29.2%) due to? In the conclusions, the authors write - "Last but not least, the treated wood samples also showed excellent anti-fouling property." - In this case, it is better to clarify in relation to which substances such properties are manifested. The authors did not use solid particles or oils, etc. in their work.

Author Response

Response to Reviewer 1 Comments

Point 1: Thank you very much for your suggestion and comment. We have corrected it in the new submitted manuscript. By calculation, the consumption of modified solution is 1487.5 g/m2. The number of layers is 3 and the drying time is 1 hour. - It's better to use kg rather than gr.

Response 1: Thank you very much for your suggestion and comment. We have corrected it in the new submitted manuscript.

Point 2: Thank you very much for your suggestion and comment. After serious consideration, we think the expression of “emulsion” is wrong, and therefore we corrected the emulsion to a solution in the newly submitted manuscript. - The authors should answer the question, what happens to the viscosity?!.

Response 2: Thank you very much for your suggestion and comment. Through measurement, the viscosity of HFSO/SiO2/V4 modifier solution is 2.23 cP.

Point 3: Line 12. "composite solution" - do I understand correctly that the mixed solution contains a second phase? Or is it a true solution, then the term composite cannot be used?.

Response 3: Thank you very much for your suggestion and comment. We have corrected it in the new submitted manuscript.

Point 4: Line 162. "the physical image", recommend removing "physical" Line 183. "absorb" - need to check the term.

Response 4: Thank you very much for your suggestion and comment. We have corrected it in the new submitted manuscript.

Point 5: Line 206. If the treated material has superhydrophobic properties, then what is the sorption of water (29.2%) due to?.

Response 5: Thank you very much for your suggestion and comment. In fact, the wood superhydrophobization treatment can not completely prevent the wood's absorption of water, but it can inhibit the wood's absorption of water. As the immersion time increases, the superhydrophobic wood inevitably absorbs some moisture. Compared with 161.3% water absorption of original wood, the water absorption of superhydrophobic wood is significantly lower.

Point 6: In the conclusions, the authors write - "Last but not least, the treated wood samples also showed excellent anti-fouling property." - In this case, it is better to clarify in relation to which substances such properties are manifested. The authors did not use solid particles or oils, etc. in their work.

Response 6: We deeply appreciate the reviewer's suggestion and comment. We have corrected it in the new submitted manuscript.

Round 3

Reviewer 1 Report

The authors took into account the reviewer's comments and made appropriate changes to the manuscript. The reviewer has no more questions for the manuscript. The manuscript may then be reviewed by the editor.